

# Developing demographic toxicity data: optimizing effort for predicting population outcomes

John D. Stark[1] and John E. Banks[2]

[1] Puyallup Research and Extension Center, Washington State University, Puyallup, WA, United States
[2] Undergraduate Research Opportunities Center, California State University, Monterey Bay, Seaside, CA, United States

## ABSTRACT

Mounting evidence suggests that population endpoints in risk assessment are far more accurate than static assessments. Complete demographic toxicity data based on full life tables are eminently useful in predicting population outcomes in many applications because they capture both lethal and sublethal effects; however, developing these life tables is extremely costly. In this study we investigated the efficiency of partial life cycle tests as a substitute for full life cycles in parameterizing population models. Life table data were developed for three species of Daphniids, *Ceriodaphnia dubia*, *Daphnia magna*, and *D. pulex*, weekly throughout the life span of these species. Population growth rates ($\lambda$) and a series of other demographic parameters generated from the complete life cycle were compared to those calculated from cumulative weeks of the life cycle in order to determine the minimum number of weeks needed to generate an accurate population projection. Results showed that for *C. dubia* and *D. pulex*, $\lambda$ values developed at >4 weeks (44.4% of the life cycle) were not significantly different from $\lambda$ developed for the full life cycle (9 weeks) of each species. For *D. magna*, $\lambda$ values developed at >7 weeks (70% of the life cycle) were not significantly different from $\lambda$ developed for the full life cycle (10 weeks). Furthermore, these cutoff points for $\lambda$ were not the same for other demographic parameters, with no clear pattern emerging. Our results indicate that for *C. dubia*, *D. magna*, and *D. pulex*, partial life tables can be used to generate population growth rates in lieu of full life tables. However, the implications of differences in cutoff points for different demographic parameters need to be investigated further.

## INTRODUCTION

A growing body of literature suggests that demography-based approaches are far more effective in determining what happens to populations subjected to stressors or disturbance than short-term acute mortality estimates (e.g., $LC_{50}$) (*Van Straalen, Schobben & De Goede, 1989*; *Forbes & Calow, 1999*; *Sibly, 1999*; *Calow, Sibly & Forbes, 2001*; *Pastorok et al., 2002*; *Stark & Banks, 2003*; *Akçakaya, Stark & Bridges, 2008*; *Barnthouse, Munns & Sorensen, 2008*). In particular, demography-based approaches can address issues such as sub-lethal effects, stage- or age-specific life history rates, and time-varying demographic processes far better than more static methods (*Stark & Banks, 2003*; *Banks et al., 2008*). However, the

Corresponding author
John E. Banks, jebanks@csumb.edu

development of demographic data is costly and time-consuming. In some toxicological risk assessments, researchers have attempted to use partial life table data instead to predict population outcomes (*Laskowski & Hopkin, 1996*; *Preston & Snell, 2001*; *Ducrot et al., 2010*). However, it is not clear how predictions from these studies incorporating partial life tables compare with studies using complete life tables. In particular, little attention has been paid to the tradeoff between accuracy and experimental effort when comparing partial vs. full life table studies. We offer here an exploration of this issue, in which population outcomes from complete life tables are compared with those developed from partial life tables for three Daphniid species. We address the issue of whether or not partial demographic data can be used in lieu of complete demographic data without loss of accuracy in projecting population outcomes. Finally, we compare outcomes for lambda vs. other demographic parameters across all species, and assess the overall potential for using reduced datasets to generate reliable population projections.

## MATERIALS AND METHODS

### Species tested

Three species of Daphniids were evaluated in this study; *Ceriodaphnia dubia* (Richard) *Daphnia pulex* (Leydig) and *D. magna* (Straus). Individuals used to develop life table data were obtained from cultures maintained at Washington State University, Puyallup Research and Extension Center. Each species was reared in reconstituted dilution water (RDW). The RDW used in this study was prepared according to a method modified from a USEPA protocol (*USEPA, 2002*) resulting in a RDW with pH 7.4–7.8, conductivity 260–320 μS, dissolved oxygen (DO) >8.0 mg/l, alkalinity of 60–70 mg/l and a hardness of 80–100 mg/l. Daphniids were maintained in an environmental chamber set with a photoperiod of 18 h: 6 h light: dark, 25.0 ± 0.1 °C, and 50.0 ± 0.1% relative humidity (RH).

The Daphniids were fed a solution consisting of a 1:1.5 mixture of yeast-cereal leaves-trout chow (YCT) and the algal species *Pseudokirchneriella subcapitata* (previously *Selenastrum capricornutum*) (Charles River Co., Wilmington, MA, USA).

### Development of life tables

Individuals (<24 h old) at or beyond the third filial (F3) generation were transferred into glass beakers containing 25 ml RDW. Founding individuals were moved to fresh RDW every other day. Three batches (replicates) of 10 individuals of each species were used to develop life tables. Individual survival and the number of offspring produced were recorded daily throughout their life span. Offspring were removed daily. Life tables were developed at weekly intervals and at the end of each species life cycle. Beakers were held in an environmental chamber under the conditions listed above for colony maintenance.

Life tables were developed following the approach outlined in *Carey (1993)* and *Vargas et al. (2002)*. The following demographic parameters were determined in this study: Net Reproductive Rate ($Ro$), the per generation contribution of newborn females to the next generation, Intrinsic Birth Rate ($b$), the per capita instantaneous rate of birth in the stable population, Intrinsic Death Rate ($d$), the per capita instantaneous rate of death in the stable population, Mean Generation Time ($T$), the time required for a newborn female to replace

herself $R_0$-fold, Doubling Time ($DT$), the time required for the population to increase twofold, Intrinsic Rate of Increase ($r_m$), the rate of natural increase in a closed population, and the Finite Rate of Increase ($\lambda$), the factor by which a population increases in size from time $t$ to time $t+1$.

## Statistical analysis

The data for all of the above mentioned demographic parameters were analyzed with one-way analysis of variance (ANOVA) (SAS Institute, Cary, NC, USA) to test for differences between results for the entire life table to the partial life tables , means of each demographic parameter for the partial life tables were compared to the results from the full life table using a Dunnett's test. For each species, we compared the value of the demographic parameter of interest for each successive week to that derived from the complete life table (full life span). We thus determined the week at which the demographic parameter value was not statistically significant different ($p < 0.005$) from the value derived using the entire life span (complete life table), heretofore referred to as the "cutoff point."

## RESULTS

### C. dubia

The time at which net reproductive rate ($Ro$), birth rate ($b$), $r_m$, and $\lambda$ were not significantly different from the complete life table for *C. dubia* was four weeks (Table 1).That is, if $Ro$ was the endpoint of interest for this species, a life table would only have to be developed for four weeks to generate the same $R_0$ value stemming from a complete life table (nine weeks). For death rate ($d$) the cutoff was three weeks while the cutoff point for both generation time ($T$) and doubling time ($DT$) for *C. dubia* was five weeks. (Table 1).

### D. pulex

The cutoff point for $d$ was three weeks while the cutoff values for, $r_m$ and $\lambda$ were four weeks. Cutoff values for *Ro, b, T, and DT* were five weeks (Table 2).

### D. magna

For *D. magna*, the cutoff times for *Ro and d* were six weeks while the cutoff times for *b, T, DT, rm*, and $\lambda$ were seven weeks (Table 3).

## DISCUSSION

A number of studies have shown that incorporating demographic data into population models has the potential to improve population projections, an approach that has been widely used in conservation and ecological risk assessment (*Forbes & Calow, 2002*; *Stark & Banks, 2003*; *Stark, Vargas & Banks, 2007*; *Forbes, Calow & Sibly, 2008*; *Hommen et al., 2010*; *Forbes et al., 2011*; *Mills et al., in press*). However, demographic data are expensive to develop, and it is not clear how much data is needed to generate sufficiently accurate population endpoints. Past studies have empirically demonstrated that partial life tables may yield accurate population projections (*Van Straalen, Schobben & De Goede, 1989*; *Oli & Zinner, 2001*), but none to our knowledge have attempted to quantify the cutoff point

Stark and Banks (2016), *PeerJ*, DOI 10.7717/peerj.2067

**Table 1** *C. dubia* life table variables determined at weekly intervals.[*]

| Life table value | Week 1 | Week 2 | Week 3 | Week 4 | Week 5 | Week 6 | Week 7 | Week 8 | Week 9 |
|---|---|---|---|---|---|---|---|---|---|
| | | | | | X ± SEM | | | | |
| $Ro$ | 12.17 ± 1.25b | 50.43 ± 2.50b | 103.17 ± 7.11b | 153.83 ± 14.00b | 199.67 ± 21.15a | 228.90 ± 23.88a | 237.90 ± 22.06a | 239.40 ± 21.69a | 239.80 ± 21.7a |
| Birth rate ($b$) | 0.35 ± 0.016b | 0.33 ± 0.007b | 0.28 ± 0.003b | 0.25 ± 0.003b | 0.23 ± 0.002a | 0.21 ± 0.000a | 0.205 ± 0.002a | 0.204 ± 0.002a | 0.204 ± 0.002a |
| Death rate ($d$) | −0.06 ± 0.011b | −0.07 ± 0.004b | −0.05 ± 0.001b | −0.04 ± 0.000a | −0.03 ± 0.000a | −0.03 ± 0.000a | −0.02 ± 0.001a | −0.02 ± 0.001a | −0.02 ± 0.001a |
| Gen. time ($T$) | 6.11 ± 0.25b | 9.78 ± 0.23b | 13.99 ± 0.28b | 17.50 ± 0.46b | 20.70 ± 0.53b | 22.98 ± 0.37a | 23.88 ± 0.23a | 24.06 ± 0.17a | 24.12 ± 0.11a |
| Doubling time ($DT$) | 1.71 ± 0.12b | 1.74 ± 0.06b | 2.09 ± 0.03b | 2.41 ± 0.03b | 2.71 ± 0.02b | 2.94 ± 0.02a | 3.03 ± 0.05a | 3.05 ± 0.05a | 3.05 ± 0.05a |
| $r_m$ | 4.090 ± 0.027b | 0.331 ± 0.007b | 0.331 ± 0.004b | 0.287 ± 0.004b | 0.255 ± 0.002a | 0.236 ± 0.001a | 0.229 ± 0.004a | 0.227 ± 0.004a | 0.227 ± 0.004a |
| Lambda ($\lambda$) | 1.507 ± 0.040b | 1.494 ± 0.016b | 1.393 ± 0.006b | 1.333 ± 0.005b | 1.291 ± 0.003a | 1.266 ± 0.002a | 1.257 ± 0.005a | 1.255 ± 0.005a | 1.255 ± 0.005a |

**Notes.**

[*], Dunnett's test.

**Table 2** *D. magna* life table variables determined at weekly intervals.[*]

| Life table value | Week 1 | Week 2 | Week 3 | Week 4 | Week 5 | Week 6 | Week 7 | Week 8 | Week 9 | Week 10 |
|---|---|---|---|---|---|---|---|---|---|---|
| | | | | | X ± SD | | | | | |
| $R_o$ | 14.07 ± 0.78b | 73.13 ± 6.21b | 98.60 ± 9.37b | 129.20 ± 13.58b | 169.50 ± 18.65b | 204.13 ± 15.07b | 222.77 ± 11.81a | 238.04 ± 11.76a | 243.47 ± 12.33a | 244.34 ± 13.58a |
| Birth rate ($b$) | 0.33 ± 0.005b | 0.35 ± 0.006b | 0.32 ± 0.003b | 0.27 ± 0.003b | 0.23 ± 0.003b | 0.21 ± 0.006b | 0.20 ± 0.008b | 0.19 ± 0.007a | 0.18 ± 0.007a | 0.184 ± 0.006a |
| Death rate ($d$) | −0.05 ± 0.003b | −0.08 ± 0.003b | −0.06 ± 0.001b | −0.05 ± 0.001b | −0.03 ± 0.001b | −0.03 ± 0.002b | −0.02 ± 0.002a | −0.02 ± 0.002a | −0.02 ± 0.002a | −0.02 ± 0.001a |
| Gen. time ($T$) | 7.00 ± 0.00b | 10.09 ± 0.14b | 12.08 ± 0.12b | 15.22 ± 0.12b | 19.24 ± 0.18b | 22.58 ± 0.43b | 24.54 ± 0.84b | 26.35 ± 0.89a | 27.11 ± 0.87a | 27.25 ± 0.72a |
| Doubling time ($DT$) | 1.84 ± 0.04b | 1.63 ± 0.04b | 1.82 ± 0.02b | 2.17 ± 0.03b | 2.60 ± 0.03b | 2.94 ± 0.10b | 3.15 ± 0.14b | 3.34 ± 0.14a | 3.42 ± 0.14a | 3.44 ± 0.12a |
| $r_m$ | 0.378 ± 0.008b | 0.425 ± 0.009b | 0.380 ± 0.004b | 0.319 ± 0.005b | 0.267 ± 0.004b | 0.236 ± 0.008b | 0.220 ± 0.010b | 0.208 ± 0.009a | 0.203 ± 0.008a | 0.202 ± 0.007a |
| Lambda ($\lambda$) | 1.459 ± 0.012b | 1.530 ± 0.014b | 1.462 ± 0.006b | 1.376 ± 0.006b | 1.305 ± 0.005b | 1.266 ± 0.010b | 1.247 ± 0.010b | 1.231 ± 0.011a | 1.225 ± 0.010a | 1.224 ± 0.009a |

**Notes.**

[*], Dunnett's test.

**Table 3** *D. pulex* **life table variables determined at weekly intervals.**[*]

| Life table value | X ± SD | | | | | | | | |
|---|---|---|---|---|---|---|---|---|---|
| | Week 1 | Week 2 | Week 3 | Week 4 | Week 5 | Week 6 | Week 7 | Week 8 | Week 9 |
| $R_o$ | 8.70 ± 1.04b | 69.80 ± 7.60b | 122.83 ± 15.95b | 169.43 ± 19.00b | 201.60 ± 18.28b | 223.50 ± 13.87a | 231.40 ± 9.43a | 234.93 ± 5.30a | 237.43 ± 2.66a |
| Birth rate ($b$) | 0.31 ± 0.013b | 0.33 ± 0.005b | 0.30 ± 0.004b | 0.26 ± 0.004b | 0.24 ± 0.006b | 0.23 ± 0.008b | 0.22 ± 0.011a | 0.22 ± 0.014a | 0.22 ± 0.017a |
| Death rate ($d$) | −0.03 ± 0.009a | −0.07 ± 0.003b | −0.05 ± 0.002b | −0.04 ± 0.003b | −0.03 ± 0.003a | −0.03 ± 0.004a | −0.03 ± 0.005a | −0.03 ± 0.006a | −0.03 ± 0.007a |
| Gen. time ($T$) | 6.27 ± 0.04b | 10.54 ± 0.07b | 13.79 ± 0.26b | 16.78 ± 0.01b | 19.12 ± 0.32b | 21.03 ± 0.75a | 21.86 ± 1.22a | 22.33 ± 1.75a | 22.70 ± 2.14a |
| Doubling time ($DT$) | 2.02 ± 0.13b | 1.72 ± 0.04b | 1.99 ± 0.03b | 2.23 ± 0.05b | 2.50 ± 0.08b | 2.70 ± 0.13a | 2.75 ± 0.18a | 2.83 ± 0.23a | 2.88 ± 0.28a |
| $r_m$ | 0.344 ± 0.021b | 0.402 ± 0.00ba | 0.349 ± 0.006b | 0.306 ± 0.007b | 0.277 ± 0.009a | 0.257 ± 0.012a | 0.250 ± 0.016a | 0.246 ± 0.020a | 0.242 ± 0.023a |
| Lambda ($\lambda$) | 1.411 ± 0.030b | 1.495 ± 0.013b | 1.417 ± 0.009b | 1.357 ± 0.009b | 1.320 ± 0.012a | 1.294 ± 0.016a | 1.284 ± 0.020a | 1.278± 0.026a | 1.275 ± 0.03a |

**Notes.**

[*], Dunnett's test.

beyond which adding more life history data does not improve accuracy. To this end, in this study we sought to determine the minimum amount of time a life table needs to be developed to get a measurement of species demographic parameters that is not statistically different from data developed over the entire life span of an organism. Our results suggest that if we were only interested in population growth rate ($\lambda$), commonly used in population studies, then partial life tables can be used without sacrificing accuracy. However, the cutoff times varied among the species we evaluated, with four weeks of data collection (instead of nine weeks) sufficient for *C. dubia* and *D. pulex*, but seven weeks (instead of ten weeks) necessary for *D. magna*. Notably, these cutoff times differed not only in real time, but also in terms of the proportion of the life span of each species. That is, accurate estimates of $\lambda$ were generated from data collected for 70–80% of the lifespan of *D. magna*, whereas it took only 44% of the lifespan of *C. dubia* and *D. pulix* to generate accurate estimates for $\lambda$. Furthermore, there were no clear patterns discernible in differences among the other demographic parameters measured for the three species. For instance, accurate estimates of the death rate ($d$) were generated earlier than accurate estimates of $\lambda$. Taken together, the variable responses among parameters and among species suggests that simple predictions relating longevity or life history ecology patterns to the amount of data we need to accurately characterize population projections may not be forthcoming.

We compared the values of $\lambda$ in the current study, as this parameter is mostly commonly used as a population endpoint in disciplines such as conservation science and ecotoxicology. However, it is important to note that an underlying assumption of the calculation of $\lambda$ in life tables is that the population undergoes continuous exponential growth. This assumption may yield misleading population predictions in cases of density-dependence or time-varying per capita reproduction (e.g., *Banks et al., 2008*). More attention might be profitably paid to such contingencies; here the use of more sophisticated mathematical models may elucidate differences in life history that are driving differences in population outcomes.

### Funding
The authors received no funding for this work.

### Competing Interests
The authors declare there are no competing interests.

### Author Contributions
- John D. Stark conceived and designed the experiments, performed the experiments, analyzed the data, wrote the paper, prepared figures and/or tables, reviewed drafts of the paper.
- John E. Banks conceived and designed the experiments, analyzed the data, wrote the paper, prepared figures and/or tables, reviewed drafts of the paper.

### Data Availability
The data is reported in the manuscript's tables.

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
