# Peer review of "Developing demographic toxicity data: optimizing effort for predicting population outcomes"

_PeerJ, doi:10.7717/peerj.2067_

## Round 0.1 · original submission · Minor Revisions

I think the key issue for your revision is to address the questions on experimental methods and analyses. I thought reviewers one and two had the most important critiques. Unlike reviewer 3, I did not find the research questions to be a particular problem. However, if you can identify places to address the third reviewer's concern, it wouldn't hurt.
I'm hopeful that addressing the issues raised by the first two reviewers won't be too time consuming, and I am looking forward to your revision and being able to move ahead with the publication process.

·

Basic reporting

See annotated pdf of the manuscript for specific comments.

Basic Reporting
The paper is generally well written, but there a few items the authors should address to improve the presentation of the study. I recommend augmenting the three tables with a figure or two that visually shows some of the results and trends.

Experimental design

Experimental Design and Validity of the Findings
Counts of survival and offspring were recorded daily and then categorized weekly for statistical analysis. However, there is no mention of repeated measures. Weren’t the same experimental units (i.e., 25-ml beaker with 10 individuals) measured repeatedly over time? If so, wouldn’t the researchers need to incorporate repeated measures in their statistical analysis? Also, the counts from the experimental units for each week (partial life table data) were compared to the same experimental units for total life cycle (full life table data). What about the lack of independence with this approach? The full life table data are dependent on, and possibly confounded with, the partial life table data (i.e., weekly data). Yet, the one-way ANOVA and SNK analysis doesn’t seem to take this into account. Wouldn’t a more valid experimental design be separate populations for partial life tables and full life tables and then statistically comparing the independent populations? Please explain this more fully in the Statistical Analysis section, possibly after consulting a statistician.

Validity of the findings

See above.

·

Basic reporting

I think the manuscript is well written. It is refreshing to see this study address a key issue, time and money vs. accuracy and scientific robustness.

Experimental design

The Student-Newman-Keuls test is not the appropriate test for this study. The authors should use the Dunnett’s test because you have a control group, the end of the study when the life table is complete, compared to each week (the partial life tables). In addition the Student-Newman-Keuls test does not allow the calculation of confidence intervals which would tell the readers how much error there is in the calculations.
Also there are repeated measures in the study that does not seem to be addressed. Since the same population is being studied there will be co-linearity in the measurements. This issue should be addressed in the statistical testing.
To strengthen the paper and your argument it might be good to setup the test again and compare the population in this study to a new population and see if you get the same results. This suggestion is optional but the repeated measures need to be addressed.

Validity of the findings

No Comments

Additional comments

However the manuscript is incredibly short and very technical with little about what each parameter means in the grand scheme of things. For this article to have a bigger impact it would be nice to have more explanation of what the parameters mean and how they can be used for risk assessment etc. Keep in mind this article will have more impact if experts outside this area, especially ecotoxicologists, can understand and interpret the manuscript.
I think this manuscript should add some more about ecotoxicology and the Endangered Species Act in risk assessment of chemicals. I suggest making points about the advantages and disadvantages of each metric used in this study and make recommendations on which may be more appropriate to measure.

·

Basic reporting

Their are some minor edits, typos and wordiness. Please see submitted reviewed draft.

Experimental design

The research question needs to be more specific. It is unclear whether the researchers are trying to ask: Can partial life tables be prepared for a cost reduction in life table preparation? What is the variability in partial life table prep? Do the demographic indicators in partial life tables vary too much so that a full life table must be developed for a control?

Also need some discussion of how the estimated parameters are valuable and reasonable predictors of life table validity.

On line 64 it says that the focus is on whether the article can explain the differences in accuracy of population responses based on reduced datasets by comparing proportional differences in life spans of three Daphniid Species.

Validity of the findings

On line 64 it says that the focus is on whether the article can explain the differences in accuracy of population responses based on reduced datasets by comparing proportional differences in life spans of three Daphniid Species. This proportional element is not found in the results.

The question needs to be rewritten to say essentially, we are comparing partial life tables to full life tables.

---

## Round 0.2 · accepted · Accept

Thanks for your thorough work addressing the reviewer's comments and responding to these through rebuttal or modification of the manuscript. I think the paper is ready for publication, and accordingly, I am accepting this revision.